# A Systematic Review of Child Health and Developmental Outcomes Associated with Low Birthweight and/or Small for Gestational Age in Indigenous Children from Australia, Canada and New Zealand

**DOI:** 10.3390/ijerph182312669

**Published:** 2021-12-01

**Authors:** Madeleine Batchelor, Stephanie J. Brown, Karen Glover, Deirdre Gartland

**Affiliations:** 1Melbourne Medical School, The University of Melbourne, Melbourne 3010, Australia; 2Intergenerational Health Department, Murdoch Children’s Research Institute, Melbourne 3052, Australia; stephanie.brown@mcri.edu.au (S.J.B.); karen.glover@sahmri.com (K.G.); deirdre.gartland@mcri.edu.au (D.G.); 3Department of Pediatrics, The University of Melbourne, Melbourne 3010, Australia; 4South Australian Health and Medical Research Institute, Adelaide 5000, Australia

**Keywords:** small for gestational age, low birthweight, Indigenous, health, development, child

## Abstract

While much is known about the health implications of low birthweight for infants and adults, there is limited information about the health implications in childhood, particularly for Indigenous children. The aim of this systematic review was to assess associations between low birthweight (LBW) and/or small for gestational age (SGA) and the developmental, physical or mental health outcomes for Australian, Canadian and New Zealand Indigenous children (5–12 years), including the potential mediating role of cultural connections. The review was guided by an Aboriginal Advisory Group established to guide the Aboriginal Families Study. Four databases were investigated with pre-determined inclusion/exclusion criteria. The search identified 417 articles after independent screening by two authors. Eight studies assessing six child outcomes were included. The review identified limited evidence, although the review suggested possible links between LBW and/or SGA and childhood asthma, lower body mass index (BMI) and poorer academic performance. Links between LBW, SGA and disability, global health and developmental vulnerability were inconclusive. One study identified cultural-based resilience as protective against perinatal adversity. In summary, research on the relationship between adverse birth outcomes and Indigenous children’s health and development is limited. Further investigation and collaboration with Indigenous communities is required to drive optimised health and social services responses and equitable system reform.

## 1. Introduction

A history of colonisation, discrimination, and land and cultural dispossession has significantly affected the health and social and emotional wellbeing of Indigenous populations [1]. Inter-generational trauma, social disadvantage and stress all have considerable implications for Indigenous women throughout pregnancy. This history of colonisation, trauma, racial discrimination and social disadvantage also influences health and lifestyle behaviours, such as smoking, nutrition and alcohol consumption. Social disadvantage, limited healthcare access and the clustering and accumulation of stressful events and social health issues contribute to Indigenous women experiencing higher prevalence of under-nutrition, cigarette smoking and overall morbidity during pregnancy compared with non-Indigenous women. Each of these factors are identified risk factors for having an infant that is born with a low birthweight in Indigenous populations [2,3]. This review will discuss the causes and the consequences of low birthweight and infants being small for their gestational age and the context in which Indigenous populations face a greater burden of perinatal adversity. It will then examine the current research, assessing associations between birthweight and size for gestational age and Indigenous child health and developmental outcomes.

### 1.1. Definitions

#### First Nations Peoples

Australia’s First Nations Peoples will respectfully be referred to as Indigenous Australians, encompassing Aboriginal and Torres Strait Islander peoples. This review will also discuss Indigenous groups located in Canada and New Zealand and refer to these groups as Indigenous Canadians and Maori respectively. The review refers to Indigenous groups collectively, however we acknowledge the vast and rich cultural diversity that exists within these populations and communities.

### 1.2. Birth Outcomes

Low birthweight is classified as the first determined weight after birth being less than 2500 g [4].

Small for gestational age (SGA) refers to babies born with a birthweight less than the 10th percentile for their gestational age. It is important to make a distinction between babies born small due to constitutional factors, such as maternal height and weight (not at increased perinatal risk), and those born small due to pathological growth-restriction in utero [5]. This distinction, however, is not always made clear in research. Therefore, this review combines the terms SGA and intrauterine growth restriction.

Causes and risk factors for low birthweight vary. However, most low birthweight babies are born either preterm and/or SGA [5]. Intrauterine growth restriction occurs due to a sub-optimal intrauterine environment where foetal nutrient supply is compromised. Stressors associated with compromised foetal growth and low birthweight include maternal under and over nutrition, maternal psychological stress, exposure to stressful events and social health issues, extremes of maternal age, maternal morbidity, such as hypertension or diabetes, indoor air pollution, smoking, and alcohol or drug use during pregnancy [6,7].

Intrauterine growth restriction has the potential to impact long-term development and increase the risk of morbidity later in life [8]. While an expanding body of literature provides evidence of longer term implications of low birthweight and/or SGA for health in adult life [9], less is known about intermediate impacts on child developmental, physical and mental health outcomes or factors which may mediate the relationship between low birthweight and/or SGA and child health outcomes for Indigenous children.

Investigation into the associations between low birthweight and SGA and child health and developmental outcomes may provide evidence contributing to increased understanding of the factors that are underlying discrepancies in health outcomes between Indigenous and non-Indigenous Australian children. Existing research shows the immediate and long-term health impacts of low birthweight [9,10]. If more information is known about health consequences in childhood, strategies may be put into place to prevent or mitigate adverse outcomes and promote improved future health, particularly in Indigenous communities where health disadvantage is apparent.

### 1.3. Consequences of Low Birthweight in Indigenous Populations

Birthweight and size are important determinants of an infant’s immediate and future physical, psychological and developmental health outcomes. Potential perinatal consequences of intrauterine growth restriction and low birthweight include perinatal asphyxia, hypoglycaemia, impaired immune function and increased risk of mortality [10]. A study estimating the burden and consequences of SGA in low-middle income countries found that 21.9% of neonatal deaths were attributable to infants born SGA [10].

Biological adaptations in response to intrauterine malnutrition have a protective function in-utero. However, they have long-lasting health and developmental consequences [11]. The Barker hypothesis proposes that early life adversity, including poor nutrition and low birthweight, predisposes adults to metabolic abnormalities [9], non-communicable disease, such as diabetes, cardiovascular disease and kidney disease, and higher mortality rates [12,13].

Early life development is influenced by a range of interactions, including biological, socio-economic, political and cultural factors. Due to the social, cultural and historical determinants of health and related inequities, Indigenous populations are particularly vulnerable to the consequences of low birthweight and/or SGA [14]. Systemic racism and discrimination, high healthcare costs, remote living and cultural and language barriers impair access to appropriate healthcare in Indigenous populations and exacerbate poor health outcomes [14].

This review was undertaken to inform analyses of data collected in the Aboriginal Families Study, a prospective study investigating the health and wellbeing of Aboriginal children and their mothers living in urban, regional and remote areas of South Australia [15]. The review was conducted under the guidance of the study’s Aboriginal Advisory Group, established in 2007 under the auspice of the Aboriginal Health Council of South Australia. Members of the Aboriginal Advisory Group include senior Aboriginal community leaders with knowledge and expertise relevant to the focus of the study [15].

The objectives of this systematic review are: (i) to determine the extent to which Indigenous children born with a low birthweight and/or SGA show poorer developmental, physical and/or mental health outcomes in middle childhood (aged 5–12 years) than Indigenous children with a healthy birthweight; (ii) to determine whether access to family, cultural or community resources and strengths mediates the relationship between low birthweight and/or SGA and outcomes for Indigenous children.

## 2. Materials and Methods

### 2.1. Search Strategy

Literature searches were conducted using the following bibliographic databases: Medline, Pubmed, Scopus and Web of Science. Search terms were developed in collaboration with a research librarian. These included: (Intrauterine growth restrict* OR small for gestational age OR low birthweight OR pre-term) AND (Aborigin* OR Indigenous) AND (Australia* OR New-Zealand* OR Canada*) AND (health OR illness* OR disability* OR development* OR depression OR resilience OR strength*) AND (children OR paediatric*).

This review sought to incorporate a strengths-based approach, and included terms such as ‘resilience’ and ‘strength’ in the Search Strategy [16]. Search terms are available in Appendix A.

The last screening search was conducted on October 13, 2021. The search included studies published after January 2000 and was limited to peer-reviewed original articles published in English. Additionally, reference lists of included research articles were searched for further relevant studies.

### 2.2. Study Selection 

#### 2.2.1. Inclusion Criteria

Articles were included based on the following criteria:Original peer-reviewed research.Prospective or retrospective cohort studies and cross-sectional studies.Indigenous study populations within Australia, Canada and New Zealand.Studies reporting data on birthweight and/or size for gestational age.Studies reporting on child developmental, physical and/or mental health outcomes at age 5–12 years stratified by low birthweight or SGA.

Indigenous children in Australia, Canada and New Zealand were included, as these three countries have similar systems of universal health insurance and evidence of ongoing health disparities between Indigenous and non-Indigenous populations [17], including higher prevalence of low birthweight and SGA infants in Indigenous populations [18].

#### 2.2.2. Exclusion Criteria

The following types of studies were excluded: case studies, qualitative studies, review papers and commentaries, unpublished studies, and quantitative studies reporting data on Indigenous populations outside of Australia, Canada and New Zealand.

### 2.3. Data Extraction

Title and abstract screening of all articles initially found in the search strategy was conducted independently by two reviewers (MB, DG). Where there was uncertainty in the title and abstract screening, full texts were obtained for inclusion/exclusion using the above criteria.

Data were extracted by a single reviewer using an investigator developed REDCap database and included study and population characteristics, exposure measures and child outcomes.

### 2.4. Critical and Cultural Appraisal

Study quality was assessed by two independent reviewers (MB, DG) using the Critical Appraisal Skills Program (CASP) Study Checklist [19], which measures the validity and applicability of research and study results using a checklist of 12 questions [19]. Cultural competence of the research was assessed independently by two reviewers (MB, DG) using the Aboriginal and Torres Strait Islander Quality Appraisal Tool [20]. This tool consists of 14 questions framed from an Aboriginal and Torres Strait Islander perspective and addressing the appropriateness of research questions, community inclusion, property rights, research collection and management, use of Indigenous paradigms, strengths-based approaches, knowledge translation and capacity strengthening of Indigenous communities [20].

## 3. Results

The search identified 772 articles. Of these, 357 were duplicates, leaving 415 articles to screen the title and abstracts. Screening was conducted independently by two reviewers with discrepancies (8%, 2/24) resolved through discussion. After full text review, six articles were selected for inclusion with an additional two articles identified through reference screening. Eight articles were included in the review (see Figure 1 PRISMA Flow Diagram). PROSPERO Registration: CRD42021248769.

### 3.1. Study Characteristics

The characteristics of the eight included studies are described in Table 1. Six of these studies were undertaken in Australia, and two were undertaken in Canada. No studies were identified from New Zealand. The sample size of Indigenous children in the included studies ranged from 419 to 48, 921 Indigenous children. The total number of Indigenous children born with a low birthweight in the included studies ranged from 8 to 11,741 Indigenous children (Table 2). The mean female representation was 48.7%.

Of the eight studies, three were prospective, three involved record linkage and two studies were cross-sectional. The term ‘record linkage study’ applies specifically to the three studies using retrospective data linkage of administrative datasets (e.g., Australian Early Development Census, Northern Territory Perinatal Data Register) in their study design [22,23,26]. Two of the eight studies, both by Guthridge et al., utilised child data from the same administrative dataset collected in the Northern Territory, Australia [22,23]. The two papers by Guthridge et al. reported on two different subsets of children, aged 8 years [22] and 4–6 years [23], and assessed different child outcomes—academic performance [22] and developmental vulnerability [23].

### 3.2. Birthweight, Birth Size and Child Health Outcomes

Table 2 describes the eight studies in terms of exposure (classified birthweight, birth size for gestational age), source of exposure data, child outcome and how it was measured. Child age when outcome data were collected ranged from 5–12 years, although four studies included children outside of this age-range [21,23,27,28]. These studies were included, as the majority of the children in each study were between 5–12 years of age. Seven studies looked solely at low birthweight as the exposure, while one study looked at both low birthweight and SGA [25]. Birthweight data sources included: caregiver reports (*n* = 5) [21,24,25,27,28], and administrative records (*n* = 3) [22,23,26].

### 3.3. Child Health Outcomes Reported in Included Studies (Aim 1)

Six child health outcomes were examined across the eight studies: asthma (*n* = 2), Body Mass Index (BMI) (*n* = 2), global health rating (*n* = 2), developmental vulnerability (*n* = 2), physical or intellectual disability (*n* = 1) and academic performance (*n* = 1).

Six studies used standardised measures. These included: the global health item from the 12-Item Short Form Health Survey (SF-12) and Physical Outcome Index (POI) to assess global health rating [21,25], the National Assessment Program-Literacy and Numeracy (NAPLAN) to assess academic performance [22], and the Australian Early Development Census (AEDC) to assess developmental vulnerability [23,26]. Study designed measures were used to assess asthma [27,28] and disability [25]. Child height and weight measurements were used to calculate BMI [24,25]. No studies used measures that were specifically developed for use with Indigenous children or referenced Indigenous definitions or understandings of health and social and emotional wellbeing [29,30].

### 3.4. Asthma

Low birthweight was associated with an increased risk of asthma in Indigenous Canadian children in two Canadian studies [27,28]. Birthweight and asthma were ascertained by caregiver reports in both studies. Asthma was assessed using a study designed measure—“Have you been told by a health care professional that your child has asthma? [27]. Senthilselvan included a second item: “Has your child had an asthma attack in the last 12 months?” [28].

Chang and colleagues conducted a cross-sectional study of 48,921 Indigenous Canadian children aged 6–14 years. The authors report that children with a low birthweight had higher odds of asthma compared to children with a healthy birthweight (adjusted odds ratio (Adj.OR) 1.44, 95% confidence interval (CI) 1.15–1.80, *p* < 0.01) after adjusting for potential confounders, including sex, obesity, allergies and dwelling condition [27].

Similar findings were reported in a cross-sectional study of 6657 Indigenous Canadian children aged 0–4 and 5–11 years (the age of low birthweight children in the analysis was unclear) [28]. In this study, a low birthweight was associated with a 68% increase in the odds of asthma compared to children born at healthy birthweight (adjusted OR 1.68, 95% CI 1.03–2.74), after adjusting for child sex, allergy, physical activity, schooling and smoke-free housing.

### 3.5. Academic Performance

A single study examined academic performance, reporting decreased performance with low birthweight [22]. The retrospective cohort study used data-linkage to match 4603 Indigenous Australian children in a Northern Territory Perinatal Data Register with Northern Territory Government school enrolment information and Grade 3 NAPLAN scores (age 8 years). Guthridge and colleagues found that low birthweight was associated with a 48% increase in odds of numeracy scores below the national minimum standard (Adj.OR 1.48, 95% CI 1.11–1.98) compared to children of healthy birthweight after adjusting for sex, age at assessment, remoteness of living, maternal age, smoking and alcohol consumption during pregnancy and caregiver school education level [22]. The odds ratio for reading scores was lower, with a 24% increase in odds of performing below the national minimum reading standard in children born with a low birthweight compared to healthy birthweight (Adj.OR 1.24, 95% CI 0.92–1.67), after adjusting for sex, age at assessment, remoteness of living, maternal age, smoking and alcohol consumption during pregnancy and caregiver school education level [22].

### 3.6. Developmental Vulnerability

Two studies examined developmental vulnerability, reporting conflicting associations between low birthweight and scores on the Australian Early Development Census [23,26]. Guthridge et al., 2016 identified 1110 Australian Indigenous children in the Northern Territory through Government school enrolment AEDC data and used data linkage to determine their birthweight using the Perinatal Data Register. Low birthweight did not increase the risk of developmental vulnerability compared with children of a healthy birthweight (adjusted OR 0.70, 95% CI 0.40–1.23), after adjusting for sex, English as a second language, pre-school attendance, estimated gestational age, Apgar score, maternal age at time of birth, plurality count, maternal parity, maternal alcohol and smoking during pregnancy, remoteness of living and primary caregiver school education level [23]. Developmental vulnerability was defined based on a child being in the lowest 10% of the national AEDC population on one or more developmental domains. The five developmental domains were: physical health and wellbeing, social competence, emotional maturity, language and cognitive skills, and communication and general knowledge [23].

Strobel et al., conducted a similar population-based cohort study linking 2715 Indigenous Australian children born in Western Australia with the Australian Early Development Census data at age 5 years. Using latent class analysis, the authors reported that 57% of those born with a low birthweight were developmentally vulnerable on one or more domains at age 5 years, compared to 48.2% of Indigenous children born with a healthy birthweight (adjusted rate ratio 1.14, 95% CI 1.00–1.30), after adjusting for sex, maternal age, gravidity, socioeconomic status, and preterm birth [26].

### 3.7. Child BMI

Two studies investigating BMI identified an increased risk of reduced child BMI in those born with a low birthweight and/or SGA [24,25]. Kim and colleagues conducted a prospective longitudinal study of 1949 Indigenous Australian children and reported that low birthweight was associated with lower predicted BMI at 11 years of age for Indigenous boys and girls compared to Indigenous children with a healthy birthweight [24]. The authors reported that the results were statistically significant for girls (*p* = 0.0001), with a trend toward lower BMI for boys. However, this finding was presented graphically with unclear numeric labelling, making it difficult to assess exact findings [24]. There was no indication of whether the discrepancy in child BMI was likely to impact on child health and wellbeing.

These findings were supported in a prospective longitudinal study of 1485 Indigenous Australian children aged 5–8 years (the Longitudinal Study of Indigenous Children (LSIC)). These data comprise two cohorts of Indigenous Australian children aged 6–18 months and 3–5 years at recruitment and followed up periodically [25] (Footprints in Time—The Longitudinal Study of Indigenous Children (LSIC)) [31]. In this study, Westrupp et al. categorised perinatal adversity into: (i) moderate-to-high risk (born very pre-term (<32 weeks), very SGA and/or very low birthweight (<1500 g)); (ii) mild risk (born at 32–36 weeks, SGA, and/or birthweight 1500–2499 g); and (iii) no increased risk (born at term, not SGA, healthy birthweight). Increased perinatal risk was associated with a linear decrease in clinically assessed BMI (mild perinatal risk: ß = −0.24, 95% CI −0.38, −0.10; moderate-high perinatal risk: ß= −0.47, 95% CI −0.69,−0.26; reference: healthy weight, term birth group) [25].

### 3.8. Global Measures of Child Health and Disability

Drawing on the Longitudinal Study of Australian Children and the 419 4–9-year-old Indigenous Australian children from the B-cohort (recruitment at 0/1 year) and K-cohort (recruitment at 4/5 years), Ou et al. reported no relationship between low birthweight and the POI composite for children aged 4–9 years in Waves 1, 2 or 3 in either cohort. Additionally, the authors used the SF-36 Global Health item to measure parent/carer report of child global health (“In general, would you say your child’s health is excellent, very good, good, fair or poor?”). They found that a low birthweight was associated with a 34% decrease in the odds of parents/carers reporting that their child’s health was ‘very good’ or ‘excellent’, compared with those of a healthy birthweight in the B-cohort (OR 0.66, 95% CI 0.47–0.93). These results were not stratified based on Indigenous status, although the authors report there was no difference between Indigenous and non-Indigenous children [21]. However, this study included only 11 and 8 children born with low birthweight in the B and K cohorts, respectively, greatly limiting the capacity to draw any conclusions.

Using data from the Longitudinal Study of Indigenous Children [31], Westrupp et al. used a single item from the SF-12 to assess child global health. Caregivers indicated child disability by responding to a list of condition types, including intellectual, learning, developmental, physical, neurological or other disabilities. Children born with moderate-to-high perinatal risk (born very pre-term (<32 weeks), very SGA and/or very low birthweight (<1500 g)) were more likely to have only fair, poor or good physical health or a reported disability, compared to excellent health or no disability, although the results were not statistically significant [25].

### 3.9. Cultural and Community Strengths (Aim 2)

Only one study assessed strengths within Indigenous culture or community in their research. Drawing on data from the Longitudinal Study of Indigenous Children, Westrupp et al., used the Resilience subscale of the Strong Souls Questionnaire to assess parent/carer report of cultural resilience and support [25]. The authors report higher resilience scores were protective against poor global health and low BMI for children born across all categories of perinatal adversity. They conclude that cultural-related resilience is a protective factor for positive child health outcomes [25].

### 3.10. Critical Appraisal

Six of the eight included studies were rated as high quality and two were rated as medium quality based on the CASP (Table 3). All studies had clearly defined research questions, acknowledged and adjusted for the impact of socioeconomic confounders in their analysis and reported plausible results that were applicable to local populations. Although study quality was rated as high for the majority of studies, a number of factors need to be taken into account in interpreting study results. In six out of eight studies, birthweight and/or SGA was not the main exposure of interest, rather it was examined as a covariate [21,22,23,26,27,28]. Some studies had limited sample sizes of low birthweight and/or SGA children based within wider study sample populations. Three studies had sample sizes of less than 330 low birthweight and/or SGA Indigenous children [23,24,26], one study reported on 11 and 8 children in two cohorts [21], and one study did not report the number of Indigenous children born with a low birthweight or SGA [28]. Findings may also have been biased by misclassification due the use of study designed items to classify child health outcomes [27,28].

### 3.11. Cultural Appraisal

Ethical codes are developed for work with Indigenous communities. These codes recognise the importance of Indigenous research being community driven and led [32,33]. This includes ownership of data, control of dissemination of research findings and inclusion of Indigenous research methodologies [32,34]. The Australian studies included in this review lacked clear reporting regarding the ways in which these ethical codes were addressed. While all research was conducted in response to health disparities between Indigenous and non-Indigenous populations, no studies reported sustained input from an Indigenous advisory group or community.

Three studies reported partial Indigenous advisory input, including one study that sought advice in the research planning phase and in the final study analysis and manuscript review [25]. One study reported that the final manuscript was reviewed by community partners and the NSW Aboriginal Health and Medical Research Council Ethics Committee [24]. Strobel et al. provided the most detailed reporting of Aboriginal advisory input, including in data interpretation and review of study findings and exhibited Indigenous leadership with the involvement of two Indigenous authors [26]. Three studies did not report on collaboration with Indigenous community members [21,22,23].

There was a lack of detail provided to determine Indigenous governance across all studies. No studies reported on: capacity strengthening with Indigenous researchers/communities; agreements to protect Indigenous intellectual or cultural property; or use of Indigenous theoretical or methodological paradigms. Despite the general lack of community involvement reported, all papers made policy or practice recommendations with an aim to benefit Indigenous Australian communities.

Although the Aboriginal and Torres Strait Islander Quality Appraisal Tool was developed with Australian Indigenous communities, concepts such as Indigenous community involvement, leadership, governance and application of Indigenous paradigms are potentially applicable to other First Nations communities. Canada has similar guidelines to Australia for research involving Indigenous people [33]. Neither of the two Canadian studies reported using any of the recommended approaches for working with Indigenous communities described above [27,28]. Greater clarity and more detailed reporting of Indigenous involvement throughout the research process is needed to determine whether Indigenous research has been community driven and led.

## 4. Discussion

This review identified only eight studies which examined six child outcomes associated with low birthweight and/or SGA in Indigenous populations: asthma, BMI, global health, disability, academic performance and developmental vulnerability. Review of study findings identified evidence of potential links between low birthweight and/or SGA and childhood asthma, decreased child BMI, and poorer academic performance. Evidence of links between low birthweight, SGA and disability, global health rating and developmental vulnerability were inconclusive.

An association between low birthweight and/or SGA and decreased child BMI was the most robust finding, although this was based on only two studies. The two prospective studies had low-moderate sample sizes and used direct measures of height and weight to calculate BMI to assess this association [24,25]. These results are supported by a 2008 literature review of the growth projection of low birthweight preterm children, which identified the risk of abnormally low early childhood weight gain in non-Indigenous children [35]. Low early childhood weight can predispose children to excessive catch-up weight gain in middle childhood which increases future risk of obesity and metabolic disease in adulthood [35]. With only two studies identified examining BMI in this review, more research needs to be done to confirm any findings in relation to Indigenous children.

Similarly, the association between low birthweight and the risk of child asthma requires further investigation [27,28]. Broader literature supports an association between low birthweight and childhood respiratory disease. In a study of 180 Australian Indigenous children who were less than 5 years of age, Hall et al., 2015, found an association between low birthweight and acute respiratory illness [36]. This was supported in a review by Grant et al., 2011, which suggests that low birthweight and intrauterine growth restriction are important factors contributing to an increased acute lower respiratory tract infection (ALRTI) burden in non-Indigenous and Indigenous children under 5 years of age in New Zealand [37]. Although the study’s authors state that these perinatal risks are not explanatory factors, they do propose that a lower birthweight increases the risk of death from ALRTI in children under 5. They furthermore state that low birthweight may be an important factor contributing to the variability in ALRTI burden in the Maori population, although results were inconsistent [37]. These results imply that low birthweight and intrauterine growth restriction may convey increased risk to the respiratory and physical health of both non-Indigenous and Indigenous children. Further research is required to better understand this risk.

Overall, the associations between birthweight or size for gestational age and physical health, disability or developmental vulnerability were inconclusive. Guthridge et al., 2016, state that the effects of low birthweight on development may have been missed due to a small sample size (*n* = 1110 Indigenous children, 13% low birthweight) resulting in a lack of study power for the analyses presented [23]. Westrupp et al., 2019 utilised a relatively weak assessment of child outcomes (caregiver report on single items) and suggested that the global self-rated measures may have resulted in caregivers under-reporting child physical health problems and disability. They call for further investigation into the influence of perinatal adversity on child health and disability in Indigenous communities [25].

The association between low birthweight and reduced academic performance was investigated by a single study in this review. However their positive finding is supported by a retrospective cohort study (McEwen et al., 2018) of 6327 Indigenous Australian children aged 8–9 years (the same dataset utilised in Guthridge, 2015) [22,38]. While not looking at low birthweight or SGA specifically, the authors report that increasing birthweight percentile was associated with a linear increase in reading and numeracy performance, as measured by NAPLAN scores. The highest scores were seen in children with birthweights above the 50th percentile [38]. Such findings suggest that large numbers of Indigenous Australian children potentially could be at an educational disadvantage, however sub-optimal birthweight is just one amongst a range of factors that influence educational outcomes [38]. These studies adjust for a range of confounders, including maternal age, sex, smoking or alcohol consumption during pregnancy, caregiver education and remoteness of residence. While accounting for all potential confounders is important, albeit challenging, particularly in studies with small sample sizes, the risk of over-adjustment must also be considered as this may mask associations. Caution is required interpreting results, especially in light of contradictory findings and much of the research appearing to have been conducted by a single research group. Further research replicating findings is needed to strengthen the evidence.

Westrupp and colleagues was the only group to recognise community and cultural-based resilience and strengths in their research [25]. They identified cultural resilience as positively mediating the association between poor child outcomes and perinatal adversity (a composite of SGA, low birthweight and preterm birth) [25]. The authors suggest health services should use a more comprehensive approach that incorporates cultural-based resilience and connection to Country to promote the social and emotional wellbeing of Indigenous children. It is important that research and support services recognise the cultural and community strengths that exist in Indigenous communities, with strengths-based approaches more likely to progress the understanding of Indigenous child health and ways of fostering positive outcomes for Indigenous children and their families.

### Strengths and Limitations

This systematic review is the first to explore the relationship between low birthweight and/or SGA and the health of Indigenous children in middle childhood. The study was conducted in a research team comprised of Aboriginal and non-Aboriginal researchers, investigators and students with an extensive history of collaborative Aboriginal and Torres Strait Islander health research in partnership with Aboriginal and Torres Strait Islander communities in South Australia. Additionally, the study was initiated and overseen by a long standing Aboriginal Advisory Group.

This review examines Indigenous child health outcomes with the aim of recognising and supporting healthy and strong Indigenous families, communities and culture, into future generations [16]. The results of this systematic review will be used to inform analyses of data collected in the Aboriginal Families Study [15].

There was a paucity of research studies identified in this review investigating the association between low birthweight and/or SGA and Indigenous child health outcomes. A lack of research diversity was identified. A quarter of the studies (2/8) used the same administrative dataset [22,23]. The majority of studies identified were Australian-based (6/8), with no studies involving Maori children identified. A small number of child outcomes has been investigated. There was little research on the developmental, behavioural or physical health of Indigenous children born with a low birthweight or SGA, and no studies investigating mental health. Further research investigating the relationship between low birthweight, SGA and child mental health and wellbeing conducted in collaboration with Indigenous communities is recommended.

Much of the outcome data on child health relied on caregiver reports or retrospectively linked administrative datasets, rather than direct assessment of child outcomes. Some studies also relied on maternal/caregiver recall of infant birthweight. This has the potential to introduce bias where participant recall and/or perspectives of health may influence the results by either increasing or decreasing associations between birthweight/size and child outcomes [39].

The majority of included studies were conducted by non-Indigenous researchers and research groups taking an epidemiological approach, with limited apparent consideration of Indigenous governance or methodological paradigms. The lack of culturally tailored measures and approaches has implications for interpreting the findings. For example, measuring academic performance in Indigenous populations using population level NAPLAN data may not adequately capture Indigenous children’s knowledge, cultural strengths or experiences [8]. Studies that lack incorporation of Indigenous perspectives may have reduced engagement of Indigenous participants, increasing the risk of culturally inappropriate research and institutional and research mistrust caused by a history of systemic discrimination [8].

## 5. Conclusions

This systematic review highlights the lack of research into the relationship between infant birthweight and size for gestational age and Indigenous children’s health and development. Better understanding of the health and developmental impacts of low birthweight and/or SGA is needed to drive optimised health and social services response and guide policy interventions. Research findings may prove vital to optimising the physical, social, emotional, and developmental outcomes of Indigenous children during their formative years. Researchers and policy makers need to work in collaboration with Indigenous communities to ensure the empowerment of Indigenous voices and culturally appropriate, relevant and inclusive research practices to inform system reform and reduce health disparities [16].

## Figures and Tables

**Figure 1 ijerph-18-12669-f001:**
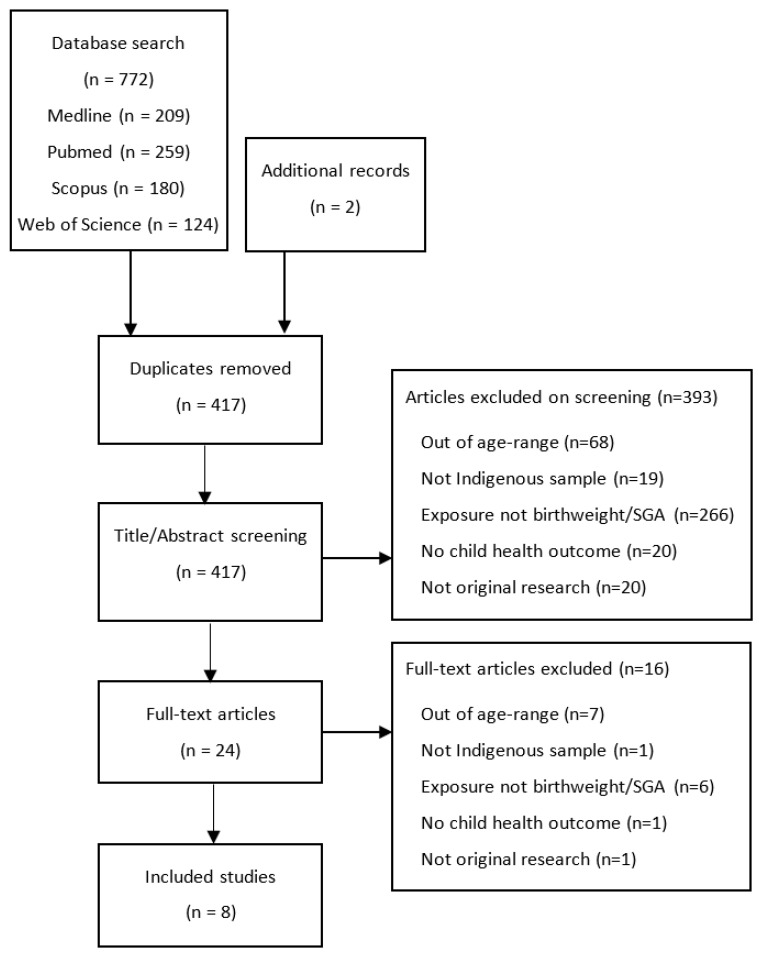
PRISMA Flow Diagram of study selection.

**Table 1 ijerph-18-12669-t001:** Summary of included papers (*n* = 8).

Paper												
First Author	Year	Country	Study Type	Setting	Sample (*n*)	Indigenous Sample (*n*)	Female (%)	Administrative Records	Caregiver Survey	Caregiver Interview	Child Survey	Clinical Assessment
Ou [21]	2011	Australia	Prospective Longitudinal	Community	10,090	419	49.5		X	X		
Guthridge [22]	2015	Australia	Record Linkage	Community	7601	4603	48.7	X				
Guthridge [23]	2016	Australia	Record Linkage	Community	1922	1110	53	X				
Kim [24]	2016	Australia	Prospective Longitudinal	Community	3418	1949	49.4		X			X
Westrupp [25]	2019	Australia	Prospective Longitudinal	Community	1483	1485	49.9		X			X
Strobel [26]	2019	Australia	Record Linkage	Community	2715	2715	50.4	X				
Chang [27]	2012	Canada	Cross-sectional	Community	48,921	48,921	48.2		X		X	
Senthilselvan [28]	2015	Canada	Cross-sectional	Community	6657	6657	40.2		X			

**Table 2 ijerph-18-12669-t002:** Classification of birthweight and size and child health outcomes reported in each paper (*n* = 8).

Study Details						Child Outcomes						
First Author	Population	Age at Outcome (Years)	Birthweight Source	Low Birthweight	Small for Gest. Age	Measures	Asthma	BMI	Global Health	Disability	Academic Performance	Developmental Vulnerability
				*n* (%)	*n* (%)							
Ou	Indigenous Australian	4–9	Caregiver	B-cohort: 11 (5)		Physical Outcome Index Composite			X			
				K-cohort: 8 (4)								
Guthridge (2015)	Indigenous Australian	8	Administrative records	580 (13)		National Assessment Program (reading and numeracy)					X	
Guthridge (2016)	Indigenous Australian	4–6	Administrative records	142 (13)		Australian Early Development Census						X
Kim	Indigenous Australian	11	Caregiver	102 (5)		Height and weight		X				
Westrupp	Indigenous Australian	5–8	Caregiver	430 (29)	430 (29)	Height, weight, SF12 global health item, caregiver report of disability		X	X	X		
Strobel	Indigenous Australian	5	Administrative records	323 (12)		Australian Early Development Census						X
Chang	Indigenous Canadian	6–14	Caregiver	11,741 (24)		Study designed measure	X					
Senthilselvan	Indigenous Canadian	0–4	Caregiver	Not reported		Study designed measure	X					
		5–11										

SF12 = 12-Item Short Form Health Survey; BMI = Body Mass Index; B-cohort = Baby cohort recruited at 0–1 years; K-cohort = Kindergarten cohort recruitment at 4–5 years.

**Table 3 ijerph-18-12669-t003:** Critical and cultural appraisal of the included studies (*n* = 8).

First Author	Critical Appraisal Quality	Cultural Appraisal Quality
Ou	High	Low
Guthridge (2015)	High	Low
Guthridge (2016)	Medium	Low
Kim	High	Low
Westrupp	High	Low
Strobel	High	Medium
Chang	High	Low
Senthilselvan	Medium	Low

## Data Availability

Not applicable.

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
