# Peer review of "A Systematic Review of Child Health and Developmental Outcomes Associated with Low Birthweight and/or Small for Gestational Age in Indigenous Children from Australia, Canada and New Zealand"

_ijerph, 2021, doi:10.3390/ijerph182312669_

Round 1

Reviewer 1 Report

This study would be systematically analyzed the health effects of SGA and Low birth weight in the Indigenous population in 3 countries. This systematic review (SR) improves the knowledge in this area. The SR is well written and the methods are clearly described. 

Here are presented my suggestion and comments to improve this SR:

  1. Lines 58,59, reference 4 is about extremely LBW, not LBW only. Please use another reference.
  2. Figure 1.  Specify the database used and relative numbers of papers in the database search section.
  3. Line 232, reference Chang et al is lacking.
  4. Table 1, The column of population source could be eliminated, because the population is the same as the previous column (country).

    Insert the paper references in the first author column

  5. Table 2, insert a legenda for B-Cohort, K-cohort, SF12, and BMI.
  6. Line 258, reference is lacking.

Reviewer 2 Report

Thank you for the opportunity to review your paper.  I enjoyed reading it.  It deals with an important set of issues.  It is well written, clear, and appropriate in length.  The methods are clearly presented and appropriate to the research question.  The results are well presented with an appropriate level of detail.  It was especially pleasing to see "Cultural Appraisal" included as a section in the results.  This is very important and it is appropriate to highlight.  The discussion is well written and is sophisticated in the way issues including confounding and causality are dealt with.  Conclusions are sound and don't over-interpret the findings.  I have no substantive criticisms. 

Author Response

Thank you for your positive comments.